# The Threshold Effect of Environmental Regulation, FDI Agglomeration, and Water Utilization Efficiency under "Double Control Actions"—An Empirical Test Based on Yangtze River Economic Belt

**Xuhui Ding** [1],*, **Ning Tang** [1] **and Juhua He** [2]

1    School of Business Administration, Hohai University, Changzhou 213022, China; tangning1998@gmail.com
2    School of Humanities, Southeast University, Nanjing 210096, China; tsinghua06hejh@126.com
*    Correspondence: dingxh@hhu.edu.cn; Tel.: +86-1381-366-8840

**Abstract:** In this study, the SE-SBM model considering undesirable outputs was used to measure the water utilization efficiency of the Yangtze River Economic Belt from 2006 to 2016, and the panel threshold model was used to estimate the impact of environmental regulation and foreign direct investment (FDI) agglomeration on water utilization efficiency. The results show that the water utilization efficiency presents a "U"-shaped trend as a whole, declines incrementally along the eastern, central, and western regions of the economic belt, and that the water utilization efficiency of the economic belt first converges and then diverges. In the estimation of the double threshold panel model, when the per capita GDP is lower than 2.635 or greater than 12.058 thousand dollars, the environmental regulation shows a significant positive effect. Otherwise, the environmental regulation barely shows a significant negative effect. FDI has not had a great impact on water resources utilization efficiency, and neither the "pollution aura" nor "pollution shelter" are significant. When the per capita GDP is lower than 2.184 or greater than 12.058 thousand dollars, FDI can significantly improve the water utilization efficiency through environmental regulation. Besides, the positive effects of technological innovation and foreign trade dependence are significant, and so are the negative effects of industrialization. Differentiated environmental regulation policies should be formulated; industrial upgrade should be promoted; innovation of water-saving and emission reduction should be strengthened in the Yangtze River Economic Belt.

**Keywords:** water utilization efficiency; environmental regulation; FDI agglomeration; threshold effect; Yangtze River Economic Belt

## 1. Introduction

The Yangtze River Economic Belt is one of the most populous and economically active regions in China and serves as an engine to promote China's economic development. Considering its long-term and high-intensity economic development, it faces serious water resource problems. Its per capita water resource availability is only about one quarter of the world average, but the demand for water resources is increasing day by day. The problem of quality-oriented water scarcity is serious. At the end of 2015, the proportion of sections in the Yangtze River basin that reached or exceeded Class III was only 73.4%. It also serves as a "water pollution refuge" for some developed countries or regions due to the irrational use of foreign capital and the transfer of industrial undertakings. The release of excess negative energy and pollution transformed the economic belt a "pollution belt" [1]. The public nature of the aquatic environment and the localization of management led local governments to act unilaterally and to participate in a so-called "race to the bottom," making the water crisis a public

management crisis (Hu, 2015) [2]. The Ministry of Water Resources issued the "13th Five Year Plan for Water Consumption Total and Intensity Dual Control Action" in 2016. This report found that the model for extensive economic growth was accompanied by a severe waste of water resources and continuous deterioration of the aquatic environment, which means environmental regulation concepts and measures need to regulate the utilization behavior of various water users. To achieve the above goals, it is important to improve water utilization efficiency and reduce unintended output. As a significant driving force for economic growth, foreign direct investment (FDI) seeks to exploit the theories of the pollution aura and pollution shelter. The advanced technologies provided by FDI, especially clean production technology, can improve the level of environmental protection, but some experts believe that investment is made by developed countries to avoid strict environmental regulation and reduce expensive environmental governance costs (Taylor, 2004) [3]. So far, there is no clear conclusion. Moreover, the theoretical and empirical results are also inconsistent with each other. More experts are beginning to focus on the factors driving water utilization efficiency, especially environmental regulation, FDI agglomeration, and water pollution transfer.

Compared with principal component analysis and cluster analysis methods, the stochastic frontier and data envelopment analysis methods are more involved in the measure and evaluation of water utilization efficiency. Traditionally, single input indicators were used to measure water utilization efficiency, such as the water consumption of ten thousand gross domestic product (GDP), which ignored the interaction between water input and other input factors (Qian et al. 2011) [4]. Guan XJ (2016), Chen HW (2014) and Chen YQ (2013) separately measured the water utilization efficiency of the Yellow River Basin, the industrial water consumption of Shandong's 17 prefecture-level cities, and household water resources utilization efficiency of Lianyungang City in Jiangsu Province by the trapezoidal fuzzy number method, the K-Means algorithm in data mining and the water price method [5–7]. Ding et al. (2018, 2019) considered undesirable output and slack variables, taking the SE-SBM model for measuring interprovincial water utilization efficiency and further proposed water ecological innovation efficiency [8,9]. Wang et al. (2018) used data envelopment analysis to measure the comprehensive efficiency, technical efficiency and scale efficiency of water resource utilization in 113 key cities for environmental protection [10]. Xu et al. (2017) used the three-stage DEA-Malmquist index method to calculate the total factor productivity of water resources in the 11 provinces and cities of the Yangtze River Economic Belt [11].

There is a study trend on environmental effects of FDI and environmental regulation, mainly focusing on the concepts of the "pollution shelter" or "pollution aura," which affect the distribution of high-water-consuming or high-pollution industries. Early literature focused on the concept of the "pollution haven," first proposed by Walter & Ugelow (1979) [12]. Copeland & Taylor (1994) formally proposed a North-South equilibrium model to explain the intrinsic relationship between international trade and pollution transfer [13]. The closely related "race to the bottom" hypothesis (Esty D.C., 1997) suggested that developing countries or regions would set lower environmental standards or relaxed environmental regulation [14]. Jiang et al. (2014) examined the impact of foreign capital entry and government regulation on water pollution and proposed the strengthening of coordinated FDI supervision [15]. The "pollution aura" represented the advanced technology and environmental protection concepts provided by FDI. Porter (1995), Letchumanan (2000), and Eskeland (2003) suggested the introduction of more advanced environmental technologies and environmental standards from multinational companies to improve the environmental conditions and resource consumption of host countries from technology spillovers [16–18]. Sheng et al. (2012) analyzed the environmental effects of FDI into scale effects, structural effects, and technical effects, and proposed that FDI was conducive to pollution reduction [19].

Most scholars found that FDI can only achieve the goals of the "pollution aura" hypothesis under a reasonable level of the environmental regulation (Manello A, 2017) [20]. The "Porter Hypothesis" also proposed that reasonable environmental regulation would lead to innovation compensation and would not hinder the entry of FDI (Ramanathan R, 2017) [21]. Luo (2017) and Shen (2012) both believed

that there was no simple "good" or "bad" solution, and there was a U-shaped, inverted U-shaped, or an N-shaped relationship [22,23]. Wu et al. (2018) empirically examined the dual threshold effect of environmental regulation and green productivity in the Yangtze River Economic Belt [24]. However, the existing research on FDI and environmental regulation mainly focus on the field of energy, and there are few studies on water resource utilization. Only Zhang (2014), Jin (2018), and Zheng (2012) investigated the technological progress index decomposition, the threshold effect estimation, and water supply enterprise performance [15,25,26]. Water resource consumption and water pollution emissions, however, have not been comprehensively considered. There are few feedback mechanisms related to FDI and government regulation and the impact of environmental regulation on the environmental effects of FDI. The proposed policies have not been closely integrated with new policies, such as the "River Chief System" and "Double Control Action." All the problems above will be answered individually in this paper. This paper would select specific areas of Yangtze River Economic Belt from the perspective of undesirable output to investigate environmental regulation, FDI agglomeration, and water utilization efficiency in order to provide policy recommendations for the construction of a green ecological Yangtze River Economic Belt.

## 2. Materials and Methods

### 2.1. Efficiency Measurement Model

The efficiency evaluation methods of water resources were involved with an analytic hierarchy process, the fuzzy comprehensive evaluation method, the projection pursuit method, the stochastic frontier analysis method, and the data envelopment analysis method (Wang, 2010) [27]. The data envelopment analysis method (DEA) [28] is a nonparametric technical efficiency analysis method based on the relative comparison between evaluated objects, which can ignore the specific functional form and standardization of data. The DEA model was proposed by American logicians Charnes et al. (1978), and Tone Kaoru (2001) proposed the slack based measure model (SBM). The SE-SBM model considering undesirable output was adopted, which can solve the problem of the radial model not containing slack variables for inefficiency measurements, and also solve the problem of differentiating the efficiency of effective DMU, while the undesirable output model also incorporates "bad" output into the measurement system [8].

The authors define water utilization efficiency as the degree to which water resources are utilized in production, household and the overall ability of using resources to create benefits (Song et al. 2018) [29]. The SBM model assumes that production systems have decision units and each decision unit can be divided into $m$ input resources $x$, $s_1$ expected output $y^g$, and $s_2$ unexpected output $y^b$. We define matrix $X$, $Y^g$, $Y^b$ as $X = [x_1, x_2, \cdots x_n]$, $Y^g = \left[y_1^g, y_2^g, \cdots y_n^g\right]$, $Y^b = \left[y_1^b, y_2^b, \cdots y_n^b\right]$, where input, expected output and unexpected output are all greater than 0. In the SE-SBM model below, $s$ is slack variable of input resource and output product and $\lambda$ is weight vector, $\rho$ is objective function of three variables $s^-$, $s^b$, $s^g$, whose value between 0 and 1, $x_{ij}$ is the $i$th input of $j$th DMU, and $y_{rj}$ is the $r$th output of $j$th DMU. When its value is 1 and $s^- = s^b = s^g$, this decision unit is valid, otherwise it is invalid or loses efficiency [30].

$$\min\rho = \frac{1 + \frac{1}{m}\sum_{i=1}^{m}\frac{s_i^-}{x_{ik}}}{1 - \frac{1}{q_1+q_2}\left(\sum_{r=1}^{q_1}\frac{s_r^{g+}}{y_{rk}^g} + \sum_{r=1}^{q_2}\frac{s_t^{b-}}{y_{rk}^b}\right)} \tag{1}$$

$$s.t. \sum_{j=1, j\neq k}^{n} x_{ij}\lambda_j - s_i^- \leq x_{ik}, \sum_{j=1, j\neq k}^{n} y_{rj}\lambda_j + s_r^{g+} \geq y_{rk}^g, \sum_{j=1, j\neq k}^{n} y_{tj}^b - s_t^{b-} \leq y_{tk}^b \tag{2}$$

$$1 - \frac{1}{q_1+q_2}\left(\sum_{r=1}^{q_1}\frac{s_r^g}{y_{rk}^g} + \sum_{r=1}^{q_2}\frac{s_r^b}{y_{rk}^b}\right) > 0, s^- > 0, s^b > 0, s^g > 0, \lambda > 0 \tag{3}$$

$$i = 1, 2, \cdots m; r = 1, 2 \cdots q; j = 1, 2 \cdots n (j \neq k) \tag{4}$$

## *2.2. Threshold Regression Model*

Different relationships of environmental regulation, FDI agglomeration, and water utilization efficiency are presented in different industries in different countries or regions. Most studies have drawn linear conclusions [31]. However, it is not accurate to describe their influence with a simple linear relationship, so a nonlinear panel threshold model should be constructed. The threshold model essentially incorporates a threshold value and constructs a piecewise function to empirically estimate the corresponding threshold effect [32]. It was initially found that environmental regulation and FDI have a nonlinear relationship, or threshold effect, on resource consumption or pollution emissions. Unless the environmental regulation intensity exceeds the specific threshold value, the "Porter hypothesis" can be achieved (Li et al. 2013) [33]. Moreover, some experts have verified the threshold effect of the "pollution refuge" or "pollution aura." In the threshold model, each threshold value is used as a critical point. Different ranges of values represent the difference in the relationship between variables. According to the number of threshold values, a single, double, or multiple threshold model can be constructed.

Hansen (2000) proposed "threshold regression" with a more rigorous statistical inference method for threshold parameter estimation and hypothesis testing [34] to circumvent the setting of and arbitrary threshold. Hansen (1999) proposed a threshold regression model considering fixed effects and simplified it using an explicit function (Equation (5)), which can be transformed into a form of dispersion and estimated using a two-step method [35]. In Equation (5), $q_{it}$ is a threshold variable, which can also be understood as a part of the independent variable, $\gamma$ is the threshold value to be estimated and $\varepsilon_{it}$ is a random disturbance term. Similarly, a multi-threshold panel regression model can be constructed by taking two threshold values (Equation (6)), where the threshold value $\gamma_1 < \gamma_2$, $(\gamma_1, \gamma_2)$ would be given. The OLS model is used to estimate the dispersion model to minimize the estimated sum of squared residuals $SSR(\gamma_1, \gamma_2)$.

$$y_{it} = \mu_i + \beta_1' x_{it} \cdot 1(q_{it} \leq \gamma) + \beta_2' x_{it} \cdot 1(q_{it} > \gamma) + \varepsilon_{it} \tag{5}$$

$$y_{it} = \mu_i + \beta_1' x_{it} \cdot 1(q_{it} \leq \gamma_1) + \beta_2' x_{it} \cdot 1(\gamma_1 < q_{it} \leq \gamma_2) + \beta_3' x_{it} \cdot 1(q_{it} > \gamma_2) + \varepsilon_{it} \tag{6}$$

## 3. Results and Discussion

### *3.1. Water Resource Utilization Efficiency*

#### 3.1.1. Data Selection and Description

The SE-SBM model considering undesirable output is proposed to measure the water resource utilization efficiency from 2001 to 2016 in Shanghai, Jiangsu, Zhejiang, Anhui, Jiangxi, Hubei, Hunan, Chongqing, Sichuan, Yunnan, Guizhou, and the other 11 provinces and cities of the Yangtze River Economic Belt. Water resources, labor, and capital should be unified into factor inputs, and economic benefits (GDP) and undesirable output (wastewater discharge) should be taken as outputs (Ma et al. 2012; Zhao et al. 2017) [36,37]. The specific data description is as follows.

Capital input. The widely used method of measuring capital stock is the perpetual inventory method was proposed by Goldsmith in 1951 [38]. Since China has not conducted a large-scale asset census, Zhang et al. (2004) proposed to select a base year estimate and then adopt the perpetual inventory method to measure the capital stock of each province and city at a constant price [39]. The data comes from the "*The Chinese Statistical Yearbook*" over the relevant years.

Labor input. The impact of labor on regional economic growth can be decomposed into labor scale, labor efficiency, and labor structure. Due to China's current status as a manufacturing power and the promotion of higher education, only the total number of urban and rural employees is selected to characterize regional labor input, and the issue of labor quality is no longer considered. The

relevant data are from "*The China Statistical Yearbook*" and "*The China Population and Employment Statistics Yearbook.*"

Total amount of water use, resource input. The statistical water consumption is not divided into three industrial scales, but into industrial water, agricultural water, domestic water, and ecological water. The use of domestic water was approximately treated as the tertiary industry water (Sun et al. 2011) [40]. The total amount is the sum of industrial water, agricultural water, and domestic water. The water consumption data of various provinces and cities are derived from "*The China Water Resources Bulletin*" and "*The China Statistical Yearbook.*"

Desirable output, real GDP. Here, the impact of price changes is removed from GDP growth. The desirable output is expressed by the real GDP of each province and city, so that the production capacity of each province and city in the period from 2000 to 2016 can be compared vertically, and the GDP growth rate of each province and city can be compared horizontally. The year 2000 is the base period, and the real GDP of each province and city are calculated based on the year 2000. The data comes from the "*The China Statistical Yearbook.*"

Undesirable output, total wastewater discharge. For the total amount of wastewater discharge, there are direct statistical indicators in the "*The China Statistical Yearbook*", "*The China Environmental Statistics Bulletin,*" and "*The China Water Resources Bulletin,*" as well as levels of lead, mercury, cadmium, arsenic, nitrogen, phosphorus, and other pollutants in wastewater. However, the specific pollutants are not subdivided here. Only the total amount of wastewater discharge is measured as the undesirable output of water resource utilization [41].

### 3.1.2. Empirical Estimation Results

The relevant data of 11 provinces in the Yangtze River Economic Belt from 2006 to 2016 was selected, and the various input and output indicators involved are as above. The SE-SBM model considering undesirable output is taken to measure the water utilization efficiency of each province and city. The results can be seen in Table 1. When using the SE-SBM model to measure the water utilization efficiency of various provinces in the economic belt, this paper sets the weight ratio of desirable output to undesirable output as 1:1. Water pollution emissions governance should be regarded as important as economic growth. In addition, the variation coefficient is also adopted to evaluate the regional difference of water utilization efficiency. The calculation formula is as followings, $CV = S/\overline{X}$, in which $CV$ is the variation coefficient, $S$ is the standard deviation of the observed variable, $\overline{X}$ is the average value [42], and the trend is shown in Figure 1.

**Table 1.** Results of interprovincial water resource utilization efficiency in the Yangtze River Economic Belt considering undesirable output.

| Province | 2006 | 2007 | 2008 | 2009 | 2010 | 2011 | 2012 | 2013 | 2014 | 2015 | 2016 |
|---|---|---|---|---|---|---|---|---|---|---|---|
| Shanghai | 0.9457 | 1.0055 | 0.9830 | 0.8429 | 0.8515 | 0.9180 | 0.9525 | 1.0111 | 1.0045 | 1.0054 | 1.0410 |
| Jiangsu | 0.8495 | 0.8413 | 0.8059 | 0.8031 | 0.8075 | 0.8090 | 0.8227 | 0.8450 | 0.8650 | 0.8801 | 0.9102 |
| Zhejiang | 0.7671 | 0.7795 | 0.7831 | 0.7834 | 0.7871 | 0.7872 | 0.7979 | 0.8131 | 0.8300 | 0.8396 | 0.8697 |
| Anhui | 0.9794 | 0.8926 | 0.8158 | 0.7863 | 0.7949 | 0.7664 | 0.7690 | 0.7692 | 0.7727 | 0.7771 | 0.8324 |
| Jiangxi | 0.7436 | 0.7413 | 0.7513 | 0.7485 | 0.7486 | 0.7333 | 0.7366 | 0.7417 | 0.7545 | 0.7537 | 0.7661 |
| Hubei | 0.7365 | 0.7389 | 0.7402 | 0.7420 | 0.7502 | 0.7486 | 0.7574 | 0.7632 | 0.7691 | 0.7725 | 0.8452 |
| Hunan | 1.0183 | 0.9682 | 0.8732 | 0.7769 | 0.7718 | 0.7724 | 0.7651 | 0.7698 | 0.7776 | 0.7886 | 0.8188 |
| Chongqing | 0.6983 | 0.7150 | 0.7103 | 0.7219 | 0.7636 | 0.7860 | 0.8110 | 0.8204 | 0.8405 | 0.8640 | 0.8023 |
| Sichuan | 0.8537 | 0.8138 | 0.7636 | 0.7737 | 0.7948 | 0.8014 | 0.8181 | 0.8162 | 0.8129 | 0.8199 | 0.8275 |
| Guizhou | 0.7092 | 0.7261 | 0.7335 | 0.7336 | 0.7435 | 0.7204 | 0.7044 | 0.7067 | 0.6776 | 0.6780 | 0.7536 |
| Yunnan | 0.7810 | 0.7802 | 0.7930 | 0.7982 | 0.7963 | 0.7141 | 0.7111 | 0.7150 | 0.7111 | 0.6998 | 0.7180 |

First, the water utilization efficiency in the period from 2006 to 2016 showed a U-shaped trend overall and continued to rise after 2011. Water consumption in most provinces was approaching the downward turning point of the U curve and the water efficiency continued to rise. In 2015, the total amount of waste water increased by 2.4%. Although it increased every year, it was far lower than

the economic growth rate, which was the most prominent in the Jiangsu, Zhejiang, and Shanghai regions. Economic growth, as shown in the "Environmental Kuznets Curve," is the ultimate means of solving environmental problems. However, the water utilization efficiency of Anhui, Hunan, Sichuan, and Yunnan provinces has declined, especially the efficiency value of Hunan in 2016, which was only 80.4%. Further study is needed to verify whether this is related to the "race to the bottom" in the local government's undertaking of industrial transfer.

Second, the overall efficiency value declined, moving from the eastern region of the economic belt (Jiangsu, Zhejiang, and Shanghai), to the central region (Anhui, Hunan, Hubei, and Jiangxi), then to the western region (Chongqing, Guizhou, Sichuan, and Yunnan). The average efficiency of the eastern, central, and western regions from 2006 to 2016 was 0.9546, 0.8659, and 0.8383, respectively. Shanghai has maintained its leading position in water utilization efficiency for ten years. In 2016, Jiangsu, Zhejiang and Shanghai also occupied the top three spots of water utilization efficiency. However, the efficiency of the provinces varied greatly in the central region in the decade. The provinces, such as Anhui and Hunan, experienced a significant decline in water utilization efficiency. The water utilization efficiency in Hubei and Jiangxi provinces has increased, but the extent is not large, while the western provinces have always had lower water utilization efficiency, although Chongqing has seen a significant improvement as a municipality [8,9].

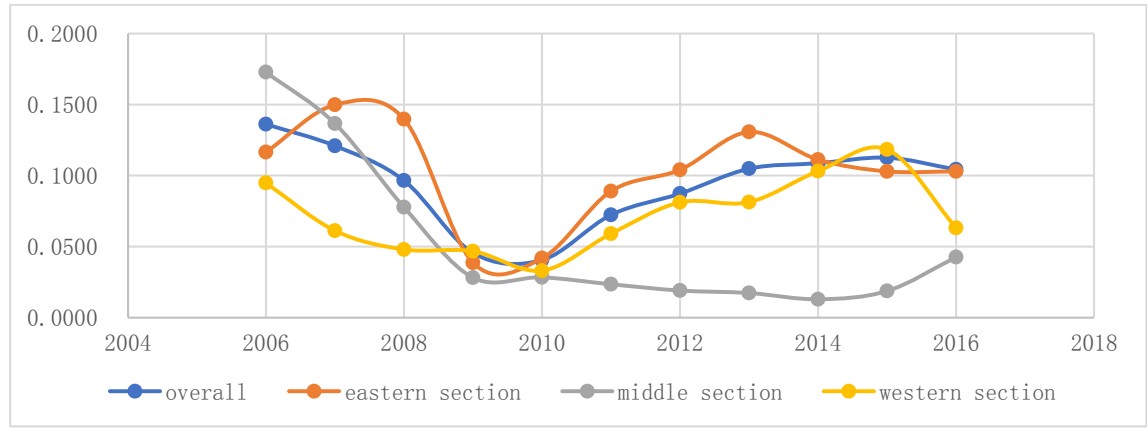

**Figure 1.** Variation-coefficient trend diagram of the overall, eastern, middle, and western sections of water utilization efficiency in the Yangtze River Economic Belt.

Third, considering the interprovincial efficiency difference, the variation coefficient of the middle region from 2006 to 2016 is the smallest. In particular, the variation coefficient is continuously decreasing, and the water utilization efficiency of the central region converged. The overall, the eastern and western regions all show U-shaped curves. Although the differences between 2006 and 2009 are constantly converging, the difference increased between 2010 and 2015, which may be due to the various development patterns as well as "Western Development" and "Rise of Central China" strategy during this period [43]. With proceeding economic development, these regions may promote some industries with high water consumption and wastewater discharge. This scenario is further explored in the Section 3.2. The variation coefficient of water efficiency's law is exactly the same as the environmental Kuznets curve. However, from 2015 to 2016, the regional convergence trend has been continuously enhanced, as a result of the technical space spillover effect in industrial agglomeration or industrial transfer.

*3.2. The Threshold Effects Regression*

3.2.1. Variable Selection and Description

The dependent variable, *Water Utilization Efficiency*. The water utilization efficiency measured by the SE-SBM model above is used, and the undesirable output is fully considered in this efficiency, distinguishing the effective decision-making units of each province and city and taking the improvement of slack variables into account. Of course, some experts adopted the water consumption intensity as utilization efficiency [44].

The core independent variable, *Environmental Regulation*. The proportion of industrial pollution control costs to industrial production added value is selected as the environmental regulation indicators, indicating the intensity of environmental regulation. The degree of industrial water pollution control also represents the strength and determination of regional water environment governance, although policies are not measured directly.

The core independent variable, *Foreign Direct Investment*. The logarithmic form of foreign direct investment is adopted to measure FDI agglomeration. FDI's resource utilization or environmental pollution in the host country is not simply good or bad [45]. It is necessary to verify the nonlinear relationship exhibited by the complex of its influence and to explore the stages of development of FDI on water utilization efficiency.

The core Independent variable, *Environmental Regulation multiply by Foreign Direct Investment*. Where the cross-item of environmental regulation and FDI are used as threshold variables, environmental regulation will indirectly affect the water utilization efficiency by affecting FDI production cost, technology spillover, and location selection.

The dependent variable of threshold regression (area system variable), *Per Capita GDP*. The logarithm of per capita GDP is used and the effect coefficient of environmental regulation, FDI, environmental regulation and FDI cross-terms are distinguished according to the node of the per capita GDP logarithm in the threshold regression.

Controlled variables. The proportion of secondary industry in the GDP is selected as the industrial structure and the number of inventions. Authorization for every 10,000 people is selected as the level of technological innovation and the per capita GDP of each province. City is selected as the level of regional development. The proportions of urban population to the total population is selected as the urbanization level. The proportions of total imports and exports as a percentage of GDP is selected as the dependence on foreign trade. The endowment of water resources is based on per capita water resource. In order to eliminate heteroscedasticity, per capita GDP and per capita water resources are both logarithmic. The data in this paper are derived from "*The China Statistical Yearbook*", and "*The China Water Resources Bulletin.*" Variables related to water utilization efficiency can be seen in Table 2.

**Table 2.** Descriptive statistics of water utilization efficiency and influence variables.

| Variable | Variable Definitions | Unit | Variable Type | Mean | Min | Max |
|---|---|---|---|---|---|---|
| effic | Water Utilization efficiency | Relative number | Dependent | 0.8001 | 0.6776 | 1.041 |
| regul | Environmental regulation | % | Independent | 0.0030 | 0.0006 | 0.0118 |
| fdi | Foreign direct investment | Hundred million | Independent | 548.3356 | 11.6684 | 2257.14 |
| pergdp | Per capita GDP | Thousand yuan | Independent | 3.7419 | 0.5787 | 11.6562 |
| indus | Proportion of second industry | % | Independent | 0.4630 | 0.2983 | 0.5635 |
| techn | Technological innovation | Pieces/10,000 | Independent | 8.1150 | 0.3540 | 42.4234 |
| city | Urbanization | % | Independent | 0.5151 | 0.2746 | 0.8960 |
| trade | Foreign trade dependence | % | Independent | 0.3319 | 0.0293 | 1.6742 |
| water | Per capita water resource | cubic meter | Independent | 1816.57 | 89.1160 | 5116.68 |

### 3.2.2. Threshold Effect Estimation

According to the efficiency analysis based on FDI and environmental regulation above, it is necessary to test the modes of single and double thresholds. From the test results in Table 3, while regul, lnfdi, and regul*lnfdi were selected as threshold variables under double threshold test, the corresponding F values are 12.2115, 16.3543, and 13.1933, respectively, all being greater than the corresponding thresholds. As the double threshold test passes the significance tests, the double threshold model will be selected in this paper [46]. In the threshold model of environmental regulation, foreign direct investment and cross-term on water utilization efficiency, each threshold value falls within the 95% confidence interval, which also indicates that the LR value is less than the critical value at the 0.05 significance level. As mentioned in the literature review, these relationships would not be unchanged simply and would change direction when the per capita GDP reaches a certain stage. These critical points may be 1.4507 ten thousand, 1.7580 ten thousand, 8.0068 ten thousand, and 9.1431 ten thousand, while approximately 0.2184, 0.2635, 1.2508, and 1.3769 thousand dollars in the international accepted fashion, which is presented in Table 4.

**Table 3.** Results of threshold conditions test and double threshold estimated value.

| Threshold Variable | Hypothetical Test | F Value | $p$ Value | Threshold Estimated | lnpergdp | Per Capita gdp (yuan) |
|---|---|---|---|---|---|---|
| regul | Single threshold | 16.2362 | 0.0010 | First threshold | 0.5642 | 1.7580 ten thousand |
| | Double threshold | 12.2115 | 0.0050 | Second threshold | 2.0803 | 8.0068 ten thousand |
| lnfdi | Single threshold | 19.7932 | 0.0000 | First threshold | 0.3721 | 1.4507 ten thousand |
| | Double threshold | 16.3543 | 0.0000 | Second threshold | 2.1230 | 9.1431 ten thousand |
| regul*lnfdi | Single threshold | 25.0760 | 0.0000 | First threshold | 0.3721 | 1.4507 ten thousand |
| | Double threshold | 13.1933 | 0.0010 | Second threshold | 2.0803 | 8.0068 ten thousand |

**Table 4.** Estimation results and tests of each threshold regression model.

| Variable | Estimated Parameter | Variable | Estimated Parameter | Variable | Estimated Parameter |
|---|---|---|---|---|---|
| regul | 8.1223(0.0664) Lnpergdp < 0.5642 −5.3649(0.1070) [0.5642, 2.0803] 22.3831(0.0019) Lnpergdp > 2.0803 | lnfdi | 0.0094(0.5563) Lnpergdp < 0.3721 −0.0068(0.6186) [0.3721, 2.1230] 0.0062(0.6721) Lnpergdp > 2.1230 | Regul*lnfdi | 2.8850(0.0257) Lnpergdp < 0.3721 −0.6631 (0.2369) [0.3721, 2.0803] 3.1695(0.0024) Lnpergdp > 2.0803 |
| lnpergdp | 0.0343 (0.4104) | lnpergdp | 0.0527 (0.2138) | lnpergdp | 0.0498 (0.1768) |
| indus | −0.4023 (0.0244) | indus | −0.2340 (0.1113) | indus | −0.3718 (0.0378) |
| techn | 0.0015 (0.0983) | techn | 0.0019 (0.0294) | techn | 0.0015 (0.0807) |
| city | −0.1411 (0.6305) | city | −0.2734 (0.3653) | city | −0.2414 (0.3976) |
| trade | 0.2261 (0.0001) | trade | 0.2196 (0.0001) | trade | 0.2282 (0.0000) |
| lnwater | 0.0113 (0.1224) | lnwater | 0.0081 (0.2535) | lnwater | 0.0070 (0.3031) |

### 3.2.3. Discuss of the Threshold Effects

It can be seen from Table 4 that the impact of environmental regulation on water utilization efficiency does have a threshold effect. It should be discussed separately according to different stages of regional economic development as follows.

(1) When the per capita GDP is less than 0.2635 thousand dollars, environmental regulation has a significant positive effect. In this situation, economic development has not yet begun. The contradiction between economic clusters and resource environment has not been fully exposed. Economic growth can largely offset the consumption of resources and pollution emissions. In 2006–2007, Jiangsu, Zhejiang, and Shanghai reached this level, so environmental regulation can play a greater role in improving water utilization efficiency. As proven in the previous literatures, "Treatment after Pollution" had been advancing for a long period to obtain the short-term development (Chen et al. 2018; Hao et al. 2018) [47,48].

(2) When the per capita GDP is between 0.2635 and 1.2058 thousand dollars, environmental regulation has a negative effect and can barely pass the significance test. In 2006–2011, the Yangtze River Economic Belt maintained a growth rate of around 10%, although economic growth is often at the expense of the environment, as in the case of the cyan bacteria crisis in Taihu Lake (Zhu 2008) [49]. Environmental regulation is difficult to implement effectively due to the competition between local governments and foreign enterprises "racing to the bottom" (Li, T. 2016) [50]. The enhancement of environmental regulation has caused a "green paradox." The necessity and effectiveness of environmental regulation policies have also been questioned at this stage. As Sinn puts forward in 2012, "good intentions do not always lead to good behavior" [51].

(3) When the Per capita GDP is more than 1.2058 thousand dollars, environmental regulation has a positive effect. When the economy shifts from the stage of high-speed development to high quality development, local governments begin to pay more attention to environmental protection. The implementation of policies such as "double control actions" and "the river chief system" forced local governments to strengthen environmental regulations (Chien, et al. 2018) [52]. However, only the regions of Jiangsu, Zhejiang, and Shanghai in the Yangtze Economic Belt reached a level of more than 1.2058 thousand in 2016, and high-water consumption and emission enterprises are beginning to relocate to the middle and upper reaches. Nevertheless, the overall water utilization efficiency of the economic belt rebounded because of technological development and industrial transformation [53].

When related to FDI, it has no significant impact on water utilization efficiency. It is not clear whether FDI provides more of a "pollution aura" or "pollution shelter" for water utilization efficiency. FDI can improve the resource allocation and utilization efficiency through advanced technology and management spillovers (Li et al., 2016) [54], but FDI and water utilization efficiency are not simply good or bad but share a complex nonlinear relationship as shown in Table 4.

(1) When the per capita GDP is less than 0.2184 thousand dollars, these regional economies are relatively backward, or industries are relatively low-end. The entry of FDI can spread cleaner production technologies, and the increase in FDI stocks can also promote the level of regional water conservation and emission reduction through business cooperation, the demonstration effect, and human capital flow. Its cross-terms show a significant positive effect, as mentioned by Javorcik (2011) [55]. Of course, different sources and features of FDI would present various effects on environmental regulation action (Fu et al. 2018) [56], while this paper considered this question in general terms and determined that it requires further scrutiny in the future.

(2) When the per capita GDP is between about 0.2184 and 1.2058 thousand dollars, the negative impact of the cross-section of FDI and environmental regulation on water utilization efficiency does not pass the significance test, which may be related to the consideration of the desirable output GDP and the undesirable output of waste water discharge together (Ding et al. 2018) [8]. With technology spillovers, there is also a squeeze in technology research and development (R&D) costs caused by increased production costs, which ultimately doesn't improve efficiency [57]. However, it has also been called the negative following cost, compared with positive innovation compensation (Zhang et al. 2011) [58].

(3) When the per capita GDP exceeds 1.2058 thousand dollars, the cross-term has a significant positive effect on water utilization efficiency. These regions (Jiangsu, Zhejiang, and Shanghai) essentially reached the level of economic development of moderately developed countries, and their

harsh environment and strong comprehensive strength enables them to attract more FDI to develop green environmental protection and high-tech industries [59]. Large-scale FDI can also accelerate its technological innovation and improve water utilization efficiency. FDI has no significant effect under the double threshold model, but environmental regulation can have a significant impact on water utilization efficiency through FDI.

As shown in Table 4, it can be first seen that the industrialization process has a significant negative effect, while the structure of industry evolves from primary industries to secondary industries and then to tertiary industries. The initial and intermediate stages of the industrialization process also mark the increase of water consumption and pollution [60]. Second, technological innovation has a significant positive effect, and future improvements in efficiency will inevitably depend on the innovation and application of advanced technologies. The number of patents per 10,000 people in Jiangsu, Zhejiang, and Shanghai is significantly higher, and this contributes to innovation in ecological and greening technology. Third, foreign trade dependence has a significant positive effect, which is inseparable from the transformation and upgrading of export-oriented enterprises in the Yangtze River Economic Belt. The dependence on foreign trade also decreased and the structure has been gradually optimized. This is also related to the increase in the import of agricultural products on the national macro level. Finally, per capita GDP, urbanization, and per capita water resources have no significant impact on water utilization efficiency.

## 4. Conclusions

In view of promoting the ecological development of Yangtze River Economic Belt, this paper adopted the SE-SBM model to measure the water utilization efficiency from 2006–2016 and adopted the panel threshold model for estimating the impact of environmental regulation and FDI agglomeration on water resource utilization efficiency. The results showed that the water utilization efficiency was generally a U-shaped trend. After 2011, the water utilization efficiency continued to increase and exceeded the average level in 2006. The water utilization efficiency went down, moving from the eastern region of the economic belt, to the central region, and to the western region. The water utilization efficiency of Jiangsu, Zhejiang, and Shanghai was relatively high. The water utilization efficiency of the whole, middle, eastern, and western sections all showed first convergent and then divergent trends. In the estimation of the double threshold panel model, when the per capita GDP was less than 2.635 thousand or more than 12.058 thousand dollars, the environmental regulation showed a significant positive effect. Otherwise, the environmental regulation showed a barely significant negative effect. Under the double threshold model, FDI has not had a significant impact on water utilization efficiency. The effect of the "pollution aura" or more "contamination evacuation" is still unclear. When the per capita GDP was less than 2.184 thousand or more than 12.058 dollars, FDI had a significant positive effect on water utilization efficiency. However, when the per capita GDP was between these two, the negative impact of FDI on water utilization efficiency did not pass the significance test. In terms of control variables, the positive effects of technological innovation and foreign trade dependence were significant, and the negative effects of industrialization were significant.

Therefore, the study recommends that the following measures should be taken. Local governments should strictly control the total water resource consumption, intensity and water pollution discharge, improve the water resource reuse rate, and optimize water resource allocation. Environmental regulation tools should be also selected appropriately and differentially in various stages of economic development because environmental regulation would have a positive effect while economic development at a certain level. The central and western regions should strictly limit "dirty industry" with high water consumption and high emissions and guide foreign investment into green environmental protection or high-technology industries in order not to become "the pollution shelter." Moreover, the government should promote the transformation and upgrade of the Yangtze River Economic Belt industry, eliminate backward production capacity, excess capacity and low-end manufacturing, and provide financial or tax support for water-saving and emission reduction

technological transformation. In order to avoid the creation of "pollution shelters," a new synergy mechanism and a reasonable compensation mechanism should be established among these provinces, along with the industry joint layout and gradient transfer.

**Author Contributions:** X.D. set the concept, analyzed the data and wrote the main manuscript text; N.T., carried out the investigation and curated the data; J.H., contributed resources tools and complemented visualization; N.T., and J.H., reviewed and edited the manuscript.

**Funding:** This research was supported by the Fundamental Research Funds for the Central Universities (2018B23614), the China Postdoctoral Science Foundation (2017M621622), and Jiangsu Social Science Fund (18GLC002)

**Conflicts of Interest:** The authors declare no conflict of interest.

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
