# Peer review of "The Threshold Effect of Environmental Regulation, FDI Agglomeration, and Water Utilization Efficiency under “Double Control Actions”—An Empirical Test Based on Yangtze River Economic Belt"

_water, doi:10.3390/w11030452_

Round 1
Reviewer 1 Report
This article explains how the use of SE-SBM model has been efficient in estimating the water use efficiency in the Yangtze River Economic Belt. The article is elaborately written and needs to go through severe structural changes before it could be accepted. Some comments are:
The title is too long. Please consider shortening it.
Line 22 Abstract: Please express GDP in $ for the international audience.
The article needs to be formatted so that it follows the guidelines of Water J. e.g. Major Headings should be Introduction, M&M, Results & Discussion & Conclusion. This article does not follow that.
Terms like FDI GDP should be elaborated the first time they appear in the text.
Bullets in the R&D section should be converted into plain text for better flow and readership.
Overall the article needs to be more succinct and crips rather than too much descriptive.
Author Response
Point 1: The title is too long. Please consider shortening it.
Response 1: I have delete some unnecessary words, such as “Research on”, “the Inter-provincial Data of the”, and finally have shorten the title as “The Threshold Effect of Environmental Regulation, FDI Agglomeration and Water Utilization Efficiency under Double Control Actions——An Empirical Test Based on Yangtze River Economic Belt”.
Point 2: Line 22 Abstract: Please express GDP in $ for the international audience.
Response 2: I have rewritten GDP in the US dollar, and also rewritten GDP in the analysis and discussion of threshold effect of Table. 3 and Table. 4. Although I adopted the Chinese currency unit to estimate the threshold effect, I have adopted the corresponding US dollar instead of “yuan” in analysis section for the international audience, such as the Line 306, 307, 316, 333, 347, 355, 363.
Point 3: The article needs to be formatted so that it follows the guidelines of Water J. e.g. Major Headings should be Introduction, M&M, Results & Discussion & Conclusion. This article does not follow that.
Response 3: I have rewritten to decompose my paper into 5 sections, separately into Introduction, Literature Review, Materials & Methods, Results & Discussion, Conclusions. I have merged the original Chapter 4 and Chapter 5, but I still reserved the Chapter “Literature Review”, which I think would explain the sources and trends of this research topic.
Point 4: Terms like FDI GDP should be elaborated the first time they appear in the text.
Response 4: I have added the full names to the abbreviation of FDI, GDP and R&D, while they appear the first time in the text, such as the Line 54, 72, 359.
Point 5: Bullets in the R&D section should be converted into plain text for better flow and readership.
Response 5: I have redefined my bullets in the Results & Discussion, such as 4.1, 4.1.1, 4.1.2, 4.2, 4.2.1, 4.2.2, 4.2.3, which for better flow and readership.
Point 6: Overall the article needs to be more succinct and crips rather than too much descriptive.
Response 6: I have streamlined my paper (except References) from about 6920 to 5720 words ,while the length of References has been increased about 600 words, for a better literature support. Moreover, I have also merged Table. 3 and Table. 4 together, and integrated Section 4 and Section 5 into Results & Discussion.

Reviewer 2 Report
See my comment in the attached file.

Author Response
Point 1: Water use efficiency is not defined appropriately either in the intro or with the literature. Given varying definitions in different disciplines, a clear definition is necessary. You also use “Water Resource Utilization Efficiency” in the text, if they are used interchangeably, please specify in a note. Otherwise, consistency in using terminologies should be followed since these terminologies have different meaning across disciplines.
Response 1: I have replaced “water use efficiency” with “water utilization efficiency” in the whole paper, in order to keep consistent and rigorous. I have given the clear definition of “water utilization efficiency”, in the Line 130-132, and also explained the application of definition in the section 4.1.
Point 2: Other terminologies like Variation-coefficient (Figure. 1) needs to be clearly presented or defined.
Response 2: I have clearly defined and explained the Variation Coefficient, with its basic calculation formula in the Line 217-220. The trend of CV has also been presented in Figure.1 and been analyzed in the Line 245-256.
Point 3: Literature review is just Okay. But you can improve this section. For example, summarize the topics in the first paragraph and using topic sentence for each paragraph. The current version seems all info is dumped together, actually this information is really related to your topic. You need summarize and synthesize all the relevant topics well. Also it’s probably because of language problems. Thus you have to improve the language significantly. See more suggestions below.
Response 3: I have improved my literature review, with summarizing the topics and using topic sentence, in order to further integrate these information, such as the Line 64-68, 69-71, 85-87, 101-102. I have also improved my language significantly, which would be answered more clearly in Response 11.
Point 4: Section 4.1, your model assumes three inputs-capital, labor and water. You need support of previous literature, and clearly present your assumption for your input-output model.
Response 4: I have added and merged more previous literature, which would support my selection of input and output variables, such as the Line 176-177. I have also clearly presented my assumption for the input-output model, such as the Line 178, 183, 189, 195, and 201.
Point 5: Section 4.1, you need an additional table showing all data with sources. This is the base you use to calculate results in table 1. The additional table can be added as an appendix, but very necessary to show the basic data info….. Well, you mentioned this in lines 258-259. These data presented in an additional table will help readers understand your study.
Response 5: I could add an additional table showing all data with sources for readers, but the table may have 121 rows (11 provinces multiply by 11 years ), as the panel data. The table may be too large, so I think I may ask for editorial comments.
Point 6: You have many unsupported claims, lines 300-305. Some results need to be discussed and compared with previous studies. Lines 287-290. Or a through discussion can work as a separate section.
Response 6: I have further rewritten the claim and added some related literature, which seemed to be unsupported and only be a speculation. The views of Lines 287-290 were regional differences in the measurement results, compared with my papers published in 2018 and 2019, marked as [8-9].
Point 7: Table 3, P values provide as much information as the 10%,5%,1% critical values. These critical values are redundant I believe. You don’t even mention these in the text.
Response 7: I have highlighted the P values instead of the 10%,5%,1% critical values in the text, which I also think more redundant, as the Lines 297-300.
Point 8: The regression model should be presented in the methods section. That is, the models used to estimate results in tables 3 and 4.
Response 8: The regression model of threshold effects has been illustrated in the formulas (2) and (3), and the estimate results in tables 3 and 4 (table 3 in the revised version) were some prior tests of threshold variables.
Point 9: The dependent variables in table 5 are not clear in the table title or in the table.
Response 9: I have further marked the dependent variables and independent variables in the Table. 3, and emphasized them in the Lines 261, 266, 271, 276, 280, 284.
Point 10: Discussion of results presented in tables 3-5 is largely missing. You need to discuss your results either with the results section or in a separate section. You also need to compare with the relevant literature; some have been presented in the literature section, while more relevant can be added. This is also related to your literature review section above.
Response 10: I have merged tables 3 and 4 into table 3, and explained the results in more details. Moreover, I also divided the discussion of table 5 (table 4 in the revised version ) into a separate section as 4.2.3. I have also added some relevant literature, such as the Lines 323, 329, 332, 337, 341, 352-353, 362, 367.
Point 11: Many statements read like government slogans, i.e., lines 474-505. Also when I read through the paper, it gives me a feeling of slogans, wording like government report. You need to have specific examples or practical wording, rather than big picture wording. Thus the paper needs through revision on language. Another issue is the grammars, punctuation, edits, etc. These needs to be correct throughout the paper. Some I mention below.
Response 11: I have rewritten this section of the policy suggestion, deleted some government slogans and reduced the length of this section. I have tried my best to propose some practical words, considering the results in Tables 1, 3,4, such as the Lines 407-421.
Point 12: Other concerns:
Lines 13-14, rephrase the sentence.
Lines 24-26, rewrite the sentence. You can split it into three.
Lines 31-33, you can use “;” to separate the three sentences.
Lines 159-160, the parentheses should be in English, rather than Chinese.
Line 184, format of citation needs to be corrected.
Punctuation need to be checked: lines 194, 253, 325, 329, … 351…
You should have carefully checked these minor issues.
Response 12: I have revised these questions above, and carefully checked the questions of language, grammar and punctuation through the whole paper. I tried my best to avoid the Chinese punctuation and Chinese expression.
Point 13: Citations and references also need to be carefully edited.
Response 13: I make sure the citations and references have been carefully edited, such as the Lines 424, 427, 435, 438.

Round 2
Reviewer 1 Report
The authors have incorporated the changes suggested. I would still recommend the authors to merge the introduction and the review of literature sections to make it a single section under Introduction.
Author Response
Point 1: The authors have incorporated the changes suggested. I would still recommend the authors to merge the introduction and the review of literature sections to make it a single section under Introduction.
Response 1: I have merged the introduction and the review of literature sections into a single section under Introduction.
Reviewer 2 Report
The presentation of the paper has been improved. Minor language issues exist, and I suggest the authors to have a native speaker check the language. Then it can be accepted for publication.
Author Response
Point 1: The presentation of the paper has been improved. Minor language issues exist, and I suggest the authors to have a native speaker check the language. Then it can be accepted for publication.
Response 1: I have improved presentation of this paper twice carefully from the beginning to the end, with the help of a native speaker in America.